# Dietary Pattern Influences Gestational Weight Gain: Results from the ProcriAr Cohort Study—São Paulo, Brazil

**DOI:** 10.3390/nu14204428

**Published:** 2022-10-21

**Authors:** Silvia Regina Dias Medici Saldiva, Adélia da Costa Pereira De Arruda Neta, Juliana Araujo Teixeira, Stela Verzinhasse Peres, Dirce Maria Lobo Marchioni, Mariana Azevedo Carvalho, Sandra Elisabete Vieira, Rossana Pulcineli Vieira Francisco

**Affiliations:** 1Departamento de Obstetrícia e Ginecologia, Faculdade de Medicina, Universidade de São Paulo, São Paulo 05403-000, Brazil; 2Department of Nutrition, Faculty of Public Health, University of São Paulo, São Paulo 01246-904, Brazil; 3Brazilian Center for Early Child Development, Insper Learning Institution, São Paulo 04546-042, Brazil; 4Department of Pediatrics, Faculty of Medicine, University of São Paulo, São Paulo 05403-000, Brazil

**Keywords:** maternal dietary patterns, body mass index, gestational weight gain, pregnant cohort study

## Abstract

The maternal pre-pregnancy body mass index (BMI) and gestational weight gain (GWG) influence maternal and infant outcomes. This study identified patterns of habitual dietary intake in 385 pregnant women in São Paulo and explored their associations with excessive weight gain (EGWG). Weight at the first visit (<14 weeks) was used as a proxy for pre-pregnancy weight. Food consumption was assessed using the 24HR method, administered twice at each gestational trimester, and dietary patterns were identified by principal component analysis. Three dietary patterns were identified: “Vegetables and Fruits,” “Western,” and “Brazilian Traditional.” Descriptive data analysis was performed using absolute and relative frequencies for each independent variable and multilevel mixed-effects logistic regression was used to analyze excessive gestational gain weight (EGWG) and dietary patterns (DP). The Brazilian Traditional dietary pattern showed a protective effect on EGWG (*p* = 0.04) and age > 35 years (*p* = 0.03), while subjects overweight at baseline had a higher probability of EGWG (*p* = 0.02), suggesting that the identification of dietary and weight inadequacies should be observed from the beginning of pregnancy, accompanied by nutritional intervention and weight monitoring throughout the gestational period to reduce risks to the mother and child’s health.

## 1. Introduction

The maternal pre-pregnancy body mass index (BMI) and subsequent gestational weight gain (GWG) are strong predictors of maternal and infant outcomes [1]. In 2009, the US Institute of Medicine (IOM), now the US National Academy of Medicine (NAM), revised the guidelines for GWG. Based on the World Health Organization’s (WHO) guidelines, they adopted specific recommendations for each pre-pregnancy BMI category to establish an acceptable range of GWG in the second and third trimesters of pregnancy [1,2]. Evidence suggests that weight gain within IOM recommendations is potentially associated with healthy fetal and maternal outcomes [3].

The prevalence of inadequate GWG varies among countries and ethnicities [4,5]. The US Pregnancy Risk Assessment Monitoring System found that 20.9% of women in the USA gained insufficient weight and 47.2% gained excessive weight in 2018. Underweight pregnancies had the highest prevalence of insufficient GWG (39.3%), while overweight and obese ones had the highest prevalence of excessive GWG (64.1% and 63.5%, respectively) [6]. Studies in Europe reveal that between 23–30% of women gain less weight during their pregnancies than the IOM recommends, while 29–48% gain more weight than the recommendations [7,8,9]. A systematic review of Brazilian studies on GWG found a higher risk of EGWG for pre-gestational overweight women [10]. 

Pre-pregnancy undernutrition and insufficient GWG increase the risk of fetal growth restriction, premature birth, low birth weight, and small for gestational age (SGA) babies, consequently increasing the risk of various morbidities and child mortality [11,12]. Pre-pregnancy excessive weight and EGWG are associated with pre-eclampsia and gestational diabetes-related complications, congenital anomalies, stillbirth, and unsuccessful breastfeeding [13,14]. Adverse outcomes for the newborns of mothers with EGWG include prematurity, the necessity for assisted ventilation, low 5-min Apgar scores, seizures, hypoglycemia, polycythemia, meconium aspiration syndrome, macrosomia, and being large for gestational age (LGA) [15]. It was also demonstrated that, in the long term, these babies were at a higher risk of obesity and chronic degenerative diseases [16]. Few studies show that obese women (>30 kg/m²) who gained insufficient weight or even lost weight during their pregnancies were at higher risk of having a newborn with a low birth weight and being SGA [17].

Several factors, such as genetics, psychological aspects, and personal behaviors, influence GWG. Diet is one of the factors that can be modified and directly or indirectly influences the health outcomes of the mother and child and possibly spans generations [18,19].

The main determinants of GWG are pre-pregnancy BMI, diet, physical activity, smoking status, level of education, and sociodemographic factors [20]. A systematic review that analyzed the role of energy and macronutrient intake (i.e., protein, fat, and carbohydrates) on the inadequacy of GWG suggested that a higher energy intake during pregnancy is associated with higher GWG. Conversely, macronutrient intake was not consistently associated with the prevalence of inadequate or excessive GWG, and these associations were comparable for pregnant women in high-income and low- and middle-income countries [21]. 

The use of dietary patterns (DPs) in nutritional studies has been proposed to overcome some of the limitations inherent in single nutrient or dietary approaches [22]. The study of DPs offers a global view of food, and these patterns are specific to the population studied and influenced by sociocultural factors and food availability [23]. International studies in cohorts of pregnant women have used dietary patterns to assess dietary quality and its influence on gestational weight gain [24,25]

Brazilian studies conducted on pregnant women from Basic Health Units showed an increased frequency of the high consumption of sugars, sweets, and fats and the low consumption of fruits, vegetables, and milk [26,27]. However, there are scarce published data in Brazil about the dietary patterns followed during the gestational period. Therefore, this study aimed to identify the dietary patterns adopted by pregnant women in São Paulo, Brazil, and their influence on the probability of excessive gestational weight gain.

## 2. Materials and Methods

### 2.1. Study Design

This study is part of the prospective cohort ProcriAr study (The Influence of Nutritional Factors and Urban Air Pollutants on Children’s Respiratory Health: A Cohort Study in Pregnant Women). Considering lung function in infants as the principal outcome, the sample size was calculated to detect a change of ≥5% in pulmonary functional parameters, with a study power of ≥80%, requiring a sample of approximately 400 pregnant women. In the same cohort, other secondary endpoints were proposed with a focus on gestational outcomes [28,29]. Data were collected between March 2011 and December 2013; 619 pregnant women were recruited from four prenatal care units in the western zone of São Paulo, Brazil. The patients were invited to participate in the study by a community health agent when they had a positive pregnancy test. An ultrasound scan was scheduled, and recruitment occurred after the first gestational ultrasound was performed. The eligibility criteria were a single fetus; Gestational Age (GA) up to 13 weeks and 6 days, confirmed by measuring the crown-rump length on the ultrasound performed in the first trimester; and absence of pre-existing chronic diseases and fetal malformations. The exclusion criteria were maternal diseases, change of address outside the recruitment area, withdrawal from participating in the project, no attendance at one of the three clinical consultations at the Clinic Hospital of the University of São Paulo, and diagnosis of miscarriage or fetal death during follow-up. Of the 619 pregnant women recruited, 6.8% did not undergo the ultrasound exam, 9.5% did not meet the inclusion criteria, 3% changed addresses, 7.3% had a miscarriage or were diagnosed without an embryo, and 11% did not attend the consultation in their third trimester. Finally, 385 pregnant women were included in the present study. The application of the inclusion and exclusion criteria is shown in Figure 1. 

### 2.2. Characteristics of the Study Group

The average age of the study group was 25.9 y (±6.3); 46.7% of the study group were primiparous, 61.5% self-declared themselves as non-white, 59% were married or partnered, 51.2% were without paid work, and 53% had >8 years of schooling. Regarding lifestyle, 83.4% of pregnant women were physically inactive and 13% were smokers. In the first trimester, (14) 3.6% of pregnant women were classified as underweight, (193) 50.1% as normal weight, (108) 28.0% as overweight, and (70) 18.2% as obese.

Information on age, education, self-reported skin color, marital status, family history of disease (mother or father), family income, housing, formal work, smoking status, alcohol consumption, ethnicity, parity, and physical activity of the participants were obtained from structured questionnaires administered face-to-face during the first consultation. Pregnant women who reported exercising were considered active. 

All participants signed informed consent forms before the study began. This study was approved by the Research Ethics Committee of São Paulo University School of Medicine (number 132/10) and the Research Ethics Committee of the Municipal Health Department of the City of São Paulo (CAAE.0205.0.162.162-10, number 430/10). 

### 2.3. Nutritional Status

The anthropometric measurements followed the recommendations of the WHO (2) and were performed by a previously trained team. Height was measured in duplicate, in in the first clinical consultation, with the participant barefoot and the head free of accessories and hairstyles and positioned in the center of the portable stadiometer Seca^®^ (UK), with accuracy of 0.1 mm and extension of 213 cm. Body weight was measured in duplicate in the three gestational trimesters with a Tanita Corporation (Tokio, Japan) portable scale with a capacity of 150 kg and variation of 0.1 kg. The weight was measured with the participants barefoot and wearing light clothes, in an upright posture, feet together, and arms extended along the body [30]. 

Weight at the first visit (<14 weeks) was used as a proxy for pre-pregnancy weight and, together with height, was used to calculate the maternal BMI (weight (kg)/height (m^2^)). BMI was categorized according to the Institute of Medicine and National Research Council/WHO [5,6] recommendations. Pregnant women younger than 19 y had their nutritional status classified according to the WHO Anthro-Plus Software, which calculates the BMI and classifies it in z-score units for age in adolescents according to the WHO reference standard, and considers the following cut-off points for its classification: underweight (z-score ≤−2), normal weight (z-score > −2–< +1), overweight (z-score ≥ +1–< +2), and obesity (z-score ≥ +2) [31]. 

The degree of incremental weight gain was calculated by subtracting the participants’ weight (kg) at the last consultation (the mean Gestational Age in this phase was 31.3 weeks) and their weight (kg) at the first consultation (mean of GA was 12.8 weeks), divided by the number of weeks in this period (kg/week) [32]. We did not use the total weight gain as an outcome because of the timing of the final weight measurement in the third trimester. The IOM-recommended weight gain (kg/week) in the second and third trimesters assumed that underweight, normal weight, overweight, and obese women should gain weight within the normal range of 0.44–0.58, 0.35–0.50, 0.23–0.33, and 0.17–0.27 kg/week, respectively. Following these recommendations, GWG is classified as excessive vs. non-excessive based on the BMI categories in the first-trimester pregnancy. 

### 2.4. Dietary Intake Assessment

The 24 h food recall method was employed to assess the dietary intake, which was administered twice in each trimester of gestation, on weekdays, weekends, and holidays. All food recalls (one in-person and the other over the phone) were applied using the Multiple Pass Method [33], which reduces dietary measurement errors by helping the interviewees remember in detail what they consumed the previous day. This stimulates the respondent to remember the food they consumed the day before through the following five steps: (1) the rapid listing of food and beverages consumed; (2) questions regarding food that is usually omitted; (3) time of consumption of food listed; (4) detailed description of food and quantities, reviewing the information about time and occasion of consumption; and (5) final review of information and probing for food that was consumed and not reported. [33]. A critical analysis of all 24 h food recalls was conducted to correct possible flaws in the description of the food, preparation of dishes, and portioning. The food and preparations were converted into grams or milliliters and standardized using tables containing the vast majority of the preparations consumed by Brazilians [34,35]. The data were entered into the Nutrition Data System for Research software 2007 version, and its associated food composition database (Nutrition Coordinating Center Food and Nutrient Database of University of Minnesota, USA) [36]. Data were analyzed for consistency, paying special attention to the measurement units of foods and preparations, and checking for the number of servings, weight, energy, and nutrient outliers. Food group intake, in grams, was adjusted for the within-person variation through the web-based statistical modeling technique Multiple Source Method (MSM) before performing principal component analysis (see below). This is a statistical method for estimating the usual nutrient and food intakes (including those episodically consumed) based on two or more short-term dietary methods, such as the 24 h recall instrument (24HR) [37]. 

### 2.5. Principal Component Analysis

The foods that were reported in the six 24 h recalls were grouped into 31 food groups for factor analysis, according to the correlation matrix and similarities in nutritional composition: beans and lentils; butter or margarine; cakes and cookies; cereal and crumbs; chocolate milk (powder); sweetened coffee; crackers; desserts and sweets; French bread; fruits; fruit smoothies and soy beverages; lean meats; fried beef, chicken, and eggs; whole milk and yogurt; mozzarella cheese; oil salad dressing; pasta dishes; pork and frankfurters, boiled potatoes and cassava; processed meat; white rice; salt; sandwich sauces; soft drinks; sweetened juices; milk and fat-reducing derivatives; sweetened tea; unsweetened juices; vegetables; vinaigrette; and wheat bread and brown rice.

Food items reported in the 24 h recall consumed by less than 5% of the population were excluded from the analysis. We excluded women with implausible total energy intakes (<600 and >6000 kcal/day). 

Dietary patterns were identified using the principal component factor analysis. The suitability of the data for factor analysis was verified using the Kaiser–Meyer–Olkin (KMO) test and the Barlett sphericity test. To identify the number of patterns to be retained, the criteria were an eigenvalue above 2.0 (Scree plot) and the interpretability of the patterns. The orthogonal Varimax rotation was also used to generate uncorrelated factors, facilitating the interpretation of the findings. The highest values of factor loadings were considered to name the patterns identified [38]. Variables with factor loadings ≥ 0.30 or ≤−0.30 were considered important for the interpretability of the factors. This procedure was initially performed with the set of six 24HRs obtained in the study, and later for each of the trimesters, that is, two 24HRs for each trimester. Finally, the patterns were named according to the main characteristics of each associated food group. For each pregnant woman, individual scores for each DP were computed and treated as continuous variables. 

### 2.6. Statistical Analyses

Descriptive data analysis was performed using absolute and relative frequencies along with and measures of central tendency and dispersion. Kruskal–Wallis test for continuous variables and chi-square test for categorical variables were implemented between EGWG (yes/no) and maternal characteristics, such as: age categories (<19, ≥19 and ≤35, and >35 years), White skin color (yes/no), education (<8 and ≥8 years), single/divorced marital status (yes/no), primiparous (yes, no), smoker or ex-smoker (yes/no), physical activity (yes/no), and nutritional status at baseline. The same tests were performed for dietary patterns and differences were considered significant at a significance level of *p* < 0.05.

To analyze the factors associated with the outcome variable, excessive weight gain (yes; no), multilevel mixed-effects binary logistic regression analysis models were applied in order to estimate odds ratios (OR) and their respective 95% confidence intervals (CI95%). Level 1 were the coefficients of the three dietary patterns for each gestational trimester, and the caloric value of the diet. Level 2 individual independent variables were BMI at baseline (BMI 25.0–29.9 kg/m^2^), age categories (<19, ≥19 and ≤35, and >35 years), and smoker or former smoker (yes/no), Level 2 was nested with level 1. To assess the parsimony of the final model, the Akaike criterion (AIC) was applied. All analyses were performed in Stata 13.0 (Stata Corp LP, College Station, TX, USA). 

## 3. Results

The sociodemographic, maternal characteristics, GA, and EGWG are presented in Table 1. The results show that 51% of the pregnant women’s GWG was above the recommendation, and the highest prevalence of excessive GWG was among overweight pregnant women (68.5%). There were no significant differences in age, ethnicity, education, income, parity, physical activity, smoking status, and GA in each trimester across participants with EGWG.

The three dietary patterns were identified as “Vegetables and Fruits,” “Western,” and “Brazilian Traditional” (Table 2). The “Vegetables and Fruits” DP was composed of salad oil and dressing, vegetables, salt, vinaigrette, fruits, and juices. The “Western” DP was composed of soft drinks, processed meat and snacks, desserts, cookies and cakes, pork and frankfurters, chocolate powder, and condiments (mayonnaise, ketchup, and mustard), while the “Brazilian Traditional” DP was composed of white rice, beans and lentils, sweetened coffee, butter and margarine, French bread, and fried beef, chicken, and eggs. Some foods such as crackers, lean meats, mozzarella cheese, boiled potatoes and cassava, fruit smoothies and soy beverages, sweetened tea, wheat bread and brown rice, reduced fat milk, pasta, whole milk and yogurts, cereal, and *farofa* (manioc flour toasted in butter, olive oil, or cooking oil) presented values below ±0.30. 

The principal component analysis of the set of the six 24 h recalls showed satisfactory adequacy, wherein the KMO was 0.5850 and the Bartlett indicator had *p* < 0.001. The explained variance was 23%. 

The % explanation of variance changed for each DP according to the pregnancy trimester. In the first trimester, the DPs comprising Vegetables/Fruits, Brazilian Traditional, and Western were found in descending order of % explanation of variance. In the second trimester, Brazilian Traditional, Vegetables/Fruits, and Western DP were found in a descending order. Lastly, in the third trimester, the order was Vegetables/Fruits, Western, and Brazilian Traditional and was similarly observed in the six 24HR sets (Appendix A).

The results of the association with maternal characteristics and the three patterns identified in this study (six 24HR sets) are presented in Table 3.

Age had significant associations with the three DPs (*p* < 0.01). The Vegetables/Fruit DP was consumed more by pregnant women who reported engaging in physical activity (*p* = 0.048). The Western DP had a significant association with single/divorced women (*p* < 0.01), primiparity (*p* < 0.01), and BMI at baseline (*p* < 0.01). The Brazilian Traditional DP had a significant association with skin color (*p* = 0.03), a low education level (*p* < 0.01), and single/divorced status (*p* < 0.01). The daily caloric value of meals varies according to the gestational trimester and nutritional status at baseline (1993–2453 kcal), where overweight and obese pregnant women presented lower values than those underweight and of normal weight (*p* < 0.001) 

The logistic regression analysis disclosed that the Brazilian Traditional DP (OR = 0.83; *p* = 0.044), age ≥ 35 y (in comparison to those aged 19–34 y (OR = 0.49; *p* = 0.034)), and smoking (OR = 0.32; *p* = 0.010) showed a protective effect on EGWG, while overweight pregnant women at baseline are at an increased risk of presenting GWG (Table 4).

## 4. Discussion

This study provides evidence that pregnant women’s DPs significantly influence their GWG. We identified three DPs and corresponding weight gain patterns in each pregnancy trimester in a cohort of Brazilian pregnant women from the ProcriAr study. The increase in GWG differed between the BMI categories identified in the first consultation, indicating that the underweight pregnant women gained more weight (0.50 kg/week) than those with obesity (0.25 kg/week). This profile is consistent with international agencies’ expectations and recommendations [1,3]. However, the overweight pregnant women at recruitment had the highest prevalence of excessive weight gain (68.5%) compared with those included in other BMI categories. This finding is relevant and suggests that pregnant women who are overweight at the beginning of pregnancy are more likely to gain excessive weight during pregnancy (*p* < 0.01) and should receive more attention from healthcare professionals during pregnancy. Studies suggest that overweight and obese pregnant women are more likely to gain excess weight than normal-weight pregnant women [6,14,39].

In Brazil, the prevalence of overweight and obesity has increased significantly in recent years due to the dietary changes characterized mainly by a typical Western diet and lower levels of physical activity [15,16,19,26,27]. Our results also suggest a high prevalence of EGWG in obese pregnant women (51%), which is considered alarming given its detrimental effects on health and pregnancy outcomes. A retrospective study in a cohort of obese American women (BMI > 30) with singleton pregnancies and live births showed that 57% of women gained weight above the recommended levels and that EGWG was associated with a higher risk of neonatal macrosomia and LGA [18].

Using the principal component analysis method, three DPs were identified for each pregnancy trimester and for all trimesters together. Notably, they are arranged differently in each of the trimesters. Between the first and second trimesters, the Vegetables/Fruits and Brazilian Traditional DPs alternate between DP1 and DP2, with the Western pattern remaining in DP3. In the third trimester, however, the Western pattern becomes DP2, and the Brazilian Traditional becomes DP3; the same pattern was found in the analysis of the six 24HR sets. The food groups with the highest prevalence of consumption were desserts and sweets (85.6%), followed by white rice (83.0%), French bread (69.0%), and beans (66.5%). The change in the position of the factors between the trimesters may be influenced by the different frequencies of consuming some food groups in the studied trimesters. For example, the fruit consumption of pregnant women at recruitment decreased from 55.7% to 48.1% and 45.7% in the second and third trimesters, respectively, while the soft drink consumption of pregnant women increased from 39.0% to 48.4% and 45.0% in the second and third trimesters, respectively. Da Mota Santana et al. (2015) reported a similar result in a longitudinal study of pregnant women in the northeastern region of Brazil, in which the consumption of some food groups such as fruits, coffee, fats, snacks, sugars, and sweets differed in the two trimesters studied [40]. The Dublin–Ireland Cohort Study of pregnant women showed that of the women who adhered to the Healthy Conscious pattern at the beginning of pregnancy, 66.9% maintained this pattern in the second trimester, while only 48.6% maintained the same in the third trimester [41]. This suggests that nutritional counseling and the promotion of a healthy diet should be initiated from the beginning of pregnancy and continue throughout, as eating habits are acquired similarly during childhood and adolescence.

The consumption of fruits and vegetables has been recommended for decades by all international health organizations for the entire population, especially for pregnant women, as they are good sources of fiber and various nutrients and generally have fewer calories than other food groups. In addition, fruits and vegetables are essential for a healthy and balanced diet. Therefore, increasing their consumption is an important public health goal [2,42]. In this study, overweight (39.8%) and obese (38.6%) pregnant women adhered more to vegetable and fruit consumption, although this finding was not statistically significant.

Adolescent pregnant women and underweight women adhered more to Western standards than other age groups. In this study, the Western pattern, composed of processed meats, desserts and sweets, soft drinks, snacks, and cookies, is consistent with previous studies that have shown that in adolescent diets, in recent years, the consumption of fast food, soft drinks, and salty snacks has increased, and the consumption of fruits and vegetables has decreased [43]. The Western DP is synonymous with ultra-processed foods rich in cholesterol, sugar, saturated fat, and sodium, which are associated with chronic non-communicable diseases and metabolic syndromes [44]. During pregnancy, consuming these foods increases the risk of excessive weight gain, diabetes, dyslipidemia, anemia, and giving birth to SGA newborns, and, thus, should be avoided [5,7,45]. Adolescent nutrition is included in several prenatal care protocols and should receive more attention and strategies should be established so that these protocols reach this age group, which is not always possible, both because of prenatal care conditions and female adolescents’ resistance to adhering to healthier DPs [46]. 

Regarding underweight pregnant women, it is important to mention that the number of this group in our study was small (*n* = 14) and may have affected the preliminary analysis of the association. However, because they have specific nutritional needs, they are usually oriented towards increased caloric and nutrient intakes and, consequently, gaining weight [1,12]. These circumstances may eventually affect their food choices, focusing on ultra-processed foods considered unfavorable for health, and may lead to adverse pregnancy outcomes [21,44]. In this study, there were no adolescents in the group of underweight pregnant women as all the participants belonged to the age group of 19–35 y. 

The Brazilian Traditional DP was followed more closely by “non-white” pregnant women who were single, had low levels of education, smoked, and were underweight. This is consistent with other studies showing a direct association between DP and sociodemographic and educational factors [27]. However, in this case, the opposite was observed because the Brazilian Traditional DP was adopted mainly by pregnant women with greater vulnerability and had a protective role against EGWG. Rice and beans are among the most consumed foods in Brazil and are considered suitable for the whole population [47]. The daily consumption of beans or other legumes, preferably at lunch and dinner, is recommended in the Brazilian food guide, especially for pregnant women, because they are rich in fiber, protein, several vitamins, and minerals, which are important for pregnancy and are iron sources [42]. Our results also show that overweight pregnant women with a low adherence to the Brazilian Traditional DP had a higher Odds ratio (OR) for EGWG. A study conducted in women in the southern region of Brazil that examined the association between DP and multimorbidity showed that a greater adherence to the Brazilian DP was associated with a lower likelihood of high scores for cardiometabolic and psychosomatic risks [48].

The results from a cohort of pregnant women in southern Africa showed that each standard deviation increase in traditional standard intake was associated with a 19% reduction in the EGWG OR [49]. The traditional DPs of southern Africa and Brazil are different; however, what stands out, in this case, is the cultural influence of the traditional diet in these regions, which provides protection against EGWG in pregnant women and is consistent with the pillars of an adequate and healthy diet, which include biological and social aspects as well as their cultural dimensions [42].

We observed that there were different proportions of adherence to dietary patterns along gestational trimesters, which could be explained according to the main symptoms and complaints that occur during pregnancy. It is known that in the first trimester, pregnant women feel more nausea and vomiting, which may cause a lack of appetite and even aversion to certain foods and food preparations, and perhaps that is why during this period there was a higher consumption of fruits and vegetables [50]. In the second trimester, hormonal changes and increased energy needs may promote an increase in appetite, a factor that may explain the greater adherence to the traditional Brazilian pattern that is composed mainly of rice and beans. In the third trimester, we can infer that the significant changes in the abdominal cavity—associated with esophageal reflux and modifications of gastric positioning—may induce the consumption of food with lower volume and higher caloric content [51]. In addition, it is possible that psychological factors associated with the proximity of childbirth may affect food patterns and consumption [52]. The clarification of such possibilities was beyond the scope of the present study and certainly deserves further investigation.

This study had some limitations. First, our database was completed nearly 10 years ago, and the results may not reflect any changes in nutritional status and dietary patterns arising from the effects of the recent COVID-19 pandemic, since the anthropometric measurements may exhibit small temporal variations because they were dependent on GA at the moment of recruitment; however, these problems were minimized, because the rate or gain of GWG per week was used instead of the total weight gain, which increased the accuracy and comparability of GWG during pregnancy [1,32,49]. Another limitation is the use of the 24HR method to report food consumption, as it is subject to errors in household measurements and respondent recall bias. For this reason, methodological techniques were used to reduce these conditions, since the 24HR was conducted on different days of the week, including weekends in each pregnancy trimester, and surveyed in different seasons of the year. Additionally, statistical methods were used to estimate the usual intakes of nutrients and foods (including those consumed episodically) at the individual level, and factor analysis using the principal components technique was employed, which is considered robust for identifying dietary patterns [33,37]. 

Our results reinforce the necessity of more intense nutrition counseling during gestation, especially in a scenario of a marked population contrast, which is shared by several countries across the globe, including Brazil. The involvement of personnel with strong nutritional backgrounds in the direct management of pre-natal, or even the use of remote tools to promote healthier eating and physical practices, are probably the best alternatives to reduce the risk of excessive gestational weight.

## 5. Conclusions

The results of this study identified three dietary patterns that exhibited different levels of adherence throughout gestation. In addition, adolescents and underweight pregnant women adhered most closely to the Western Dietary Pattern. Overweight women in early pregnancy exhibited excessive gestational weight gain, reinforcing the importance of the identification of nutritional and weight inadequacies that should be monitored from the beginning of pregnancy. Such findings confirm the necessity of nutritional interventions and weight control throughout the gestational period. We found that the Brazilian Traditional DP during pregnancy has a protective effect on excessive GWG. Investments in preserving and promoting the rich culinary histories that evolved our traditional dietary pattern should be prioritized due to its potential impact on maternal and child health.

## Figures and Tables

**Figure 1 nutrients-14-04428-f001:**
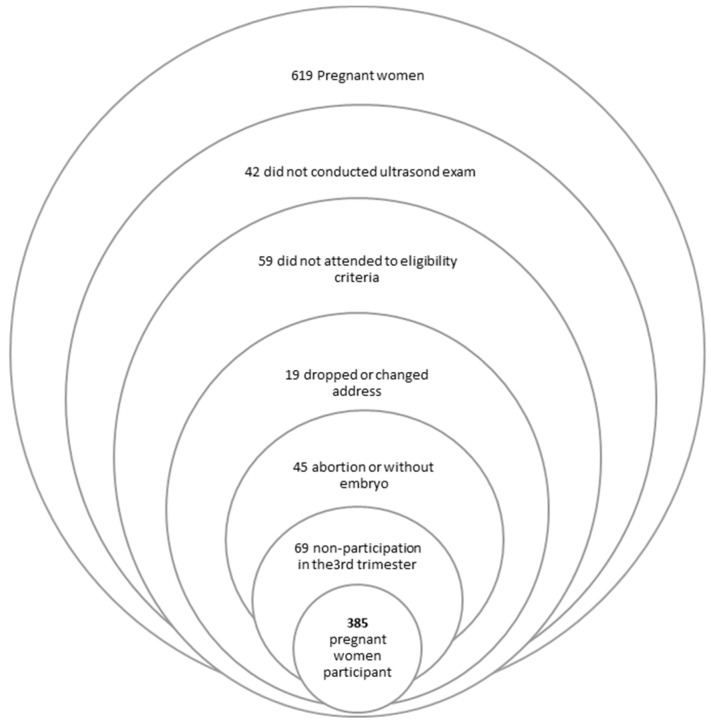
Description of the sample selection for ProcriAr study—São Paulo/Brazil, 2011–2013.

**Table 1 nutrients-14-04428-t001:** Sociodemographic, maternal characteristics, and gestational age in each trimester with respect to excessive gestational weight gain in the ProcriAr cohort, São Paulo, 2011–2013.

Maternal Characteristics	No EGWG (%)	EGWG (%)	*p*-Value
	189(49.1)	196(50.9)	
Age at enrollment (y) ^a^			0.150
<19	32(55.2)	26(44.8)	
19–34.9	135(46.4)	156(53.6)	
≥35	22(61.1)	14(38.9)	
White Skin Color ^b^			0.778
Yes	74(50))	74(50)	
No	115(48.5)	122(51.5)	
Years of schooling ^b^			0.175
<8 y	95(52.8)	85(47.2)	
≥8 y	94(45.8)	111(54.2)	
Income ^b^			0.182
≤1 Basic wage	30(57.7)	22(42.3)	
>1 Basic wage	159(47.8)	174(52.2)	
Parity ^b^			0.590
Primiparous	91(50.6)	89(49.4)	
Multiparous	98(47.8)	107(52.2)	
Marital status ^b^			0.051
Single or divorced	87(55.1)	71(45.2)	
Married/partnered	102(44.9)	125(55.1)	
Physical activity ^b^			0.226
No	162(50.5)	159(49.5)	
Yes	27(42.2)	37(57.8)	
Currently smoking ^b^			0.077
No	119(45.9)	140(54.1)	
Yes/ex-smoking	70(55.6)	56(44.4)	
BMI categories ^a^			<0.01
<18.5 kg/m^2^	9(64.3)	5(35.7)	
18.5–24.9 kg/m^2^	112(58.1)	81(41.9)	
25.0–29.9 kg/m^2^	34(31.5)	74(68.5)	
≥30 kg/m^2^	34(48.6)	36(51.4)	
Gestational Age in each trimester ^a^			
First, mean (SE)	12.3 ± 0.06	12.4 ± 0.06	0.823
Second, mean (SE)	21.7 ± 0.07	21.6 ± 0.06	0.724
Third, mean (SE)	31.7 ± 0.07	31.8 ± 0.07	0.465

^a^ Kruskal–Wallis test (continuous variables); ^b^ Mann–Whitney (categorical variables); BMI, body mass index; GA, gestational age; SE, standard of error; EGWG, excessive gestational weight gain.

**Table 2 nutrients-14-04428-t002:** Factors loadings of food groups’ characteristic to the principal dietary components identified in 24HR in pregnant women of the ProcriAr Cohort, São Paulo, 2011–2013.

Food Group	Vegetables and Fruits	Western	Traditional
Vegetables	**0.7520**	−0.1438	0.0616
Oil and salad dressing	**0.6764**	0.0841	0.2852
Fruits	**0.5037**	−0.1020	−0.1347
Sweetened juices	**0.4320**	0.0945	−0.1136
Salt	**0.4149**	0.1366	0.1774
Vinaigrette	**0.3685**	0.1246	**0.4032**
Unsweetened juices	**0.3152**	−0.0278	−0.1238
French bread	−0.2973	0.1165	**0.4346**
Butter and Margarine	−0.2826	−0.0352	**0.4305**
Milk fat-reduced	0.2722	−0.0269	−0.2680
Lean meats	0.2441	−0.1395	0.1032
Fruit smoothies and soy beverage	0.2006	−0.0021	−0.1760
White rice	0.1991	0.0669	**0.6856**
Crackers	0.1813	−0.1294	−0.0731
Sweetened tea	0.1650	−0.1273	−0.1767
Whole milk and yogurts	0.1505	0.0075	−0.1952
Beans and lentils	0.1480	−0.1531	**0.6758**
Mozzarella cheese	0.1436	0.1777	−0.0330
Cereals and farofa	−0.1394	−0.0001	−0.1937
Pork and frankfurters	0.1393	**0.4079**	0.2532
Soft drinks	−0.1185	**0.6941**	0.1113
Wheat bread and brown rice	0.1135	−0.1457	−0.2592
Pasta	0.1042	0.1802	−0.1113
Condiments	−0.1013	**0.3320**	−0.0476
Sweetened coffee	−0.0882	**−0.4299**	0.2607
Desserts and sweets	0.0486	**0.5700**	−0.0923
Processed meat and snacks	−0.0336	**0.6481**	−0.0620
Fried beef, chicken, and eggs	0.0332	0.0849	**0.3496**
Potato and cassava boiled	0.0286	0.1952	0.1108
Chocolate powder	−0.0226	**0.3654**	−0.2334
Cookies and cakes	−0.0132	**0.4723**	0.0638
Eigenvalue	2.47	2.35	2.28
Explained variance (%)	7.99	7.59	7.39

Food groups presented had factor loadings of −0.3 or 0.3 and were, therefore, used to describe each dietary pattern.

**Table 3 nutrients-14-04428-t003:** Maternal sociodemographic, BMI, and EGWG according to the three dietary patterns identified in the 24HR sets, ProcriAr, São Paulo 2011–2013.

Mothers’ Characteristics	Set of R24h								
	Vegetables/Fruits				Western		Traditional		
	Mean	SE	*p*	Mean	SE	*p*	Mean	SE	*p*
**Age ^a^**			**0.007**			**<0.001**			**0.002**
< 19 years	−0.332	0.124		0.475	0.127		0.383	0.127	
19 a 35 years	0.044	0.060		−0.044	0.057		−0.051	0.059	
> 35 years	0.172	0.117		−0.407	0.148		−0.206	0.148	
**White Skin Color ^b^**			0.632			0.145			**0.041**
Yes	0.018	0.077		0.114	0.089		−0.159	0.082	
No	−0.012	0.067		−0.071	0.061		0.099	0.064	
**Schooling ^b^**			0.054			0.603			**0.003**
< 8 years	−0.123	0.073		−0.028	0.076		0.134	0.067	
≥ 8 years	0.108	0.073		0.026	0.069		−0.118	0.074	
**Marital Status ^b^**			0.095			**0.001**			**0.004**
Single or divorced	−0.097	0.074		0.194	0.077		0.155	0.082	
Married /partnered	0.067	0.069		−0.135	0.066		−0.108	0.064	
**Income ^b^**			0.427			0.141			0.940
<1 Basic wage	−0.124	0.128		−0.153	0.143		0.096	0.162	
≥1 Basic wage	0.019	0.055		0.024	0.054		−0.001	0.053	
**Primiparous ^b^**			0.068			**<0.001**			0.599
No	0.078	0.071		−0.179	0.068		0.009	0.074	
Yes	−0.088	0.072		0.204	0.074		−0.008	0.075	
**Practice physical activity ^b^**			**0.048**			0.858			0.293
No	−0.043	0.055		−0.002	0.057		0.023	0.054	
Yes	0.218	0.125		0.009	0.114		−0.115	0.143	
**Smoking ^b^**			0.426			0.186			**0.041**
No	0.036	0.060		−0.057	0.059		−0.072	0.060	
Yes	−0.076	0.094		0.117	0.096		0.149	0.092	
**BMI at 1st trimester ^a^**			0.542			**0.021**			**0.013**
<18.5 kg/m²	−0.092	0.271		0.155	0.308		0.267	0.226	
18.5–24.9 kg/m²	−0.044	0.069		0.131	0.070		0.121	0.075	
25.0–29.9 kg/m²	0.098	0.098		−0.141	0.102		−0.086	0.088	
≥ 30.0 kg/m²	−0.012	0.127		−0.175	0.107		−0.253	0.116	
**Excessive weight gain ^b^**			0.328			0.869			0.249
No	−0.052	0.069		−0.013	0.069		0.075	0.075	
Yes	0.051	0.074		0.013	0.074		−0.072	0.069	

^a^ Kruskal–Wallis test (continuous variables); ^b^ Mann–Whitney; SE, standard error; BMI, Body mass index; EGWG, excessive gestational body weight; 24HR, 24 hr recall instrument.

**Table 4 nutrients-14-04428-t004:** Results of mixed-effects logistic regression relating EGWG to maternal characteristics and dietary pattern. ProcriAr, São Paulo 2011–2013.

Excessive Weight Gain	β (SE)	OR	*p*	L95%CI	H95%CI
Brazilian Traditional	−0.192 (0.10)	0.83	**0.044**	0.69	0.99
Daily caloric intake (kcal)	0.0004 (0.0001)	1.00	**0.010**	1.00	1.00
Underweight/normal weight		1			
Overweight at baseline	0.627 (0.27)	1.87	**0.022**	1.10	3.20
Obese at baseline	−0.112 (0.32)	0.89	0.728	0.48	1.68
Age < 19 years	−0.671 (0.41)	0.51	0.103	0.23	1.15
Age 19−34.9 years		1			
Age ≥ 35 years	−0.704 (0.33)	0.49	**0.034**	0.26	0.95
Non-smoking		1			
Smoking	−0.917 (0.36)	0.32	**0.010**	0.20	0.80

Odds Ratios: L95%CI—lower 95% confidence interval; H95%CI—higher 95% confidence interval; model Fit—Log-likelihood = −681.55 and AIC = 1379.10.

## Data Availability

Not applicable.

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
