# Peer review of "Dietary Pattern Influences Gestational Weight Gain: Results from the ProcriAr Cohort Study—São Paulo, Brazil"

_nutrients, 2022, doi:10.3390/nu14204428_

Round 1

Reviewer 1 Report

The paper investigates the association of dietary patterns with gestational weight gain (GWG) among 385 pregnant women living in São Paulo, Brazil. Based on 24HR food recall survey, 3 major dietary patterns (Vegetables and Fruits, Western, and Brazilian Traditional) were identified by PCA analysis. The authors found that Brazilian Traditional dietary pattern showed a protective effect on excessive gestational weight gain (EGWG) while overweight at baseline had a higher risk for EGWG. The finding from paper is interesting and provides an important insight on the nutritional intervention and weight monitoring during pregnancy. However, there are a few concerns that should be addressed.

Title: It is suggested to change to a more informative title with the major finding of the paper.

Line 19, linear regression analysis is not described in the method. Please add it.

Line 20, the analysis of continuous dietary pattern score is not described in the method. Please add it.

Line 72, protein, fat and carbohydrate are considered as macronutrients rather than micronutrients.

Line 212-219, the paragraph that describes the method of multilevel mixed effects logistic regression is confusing. Please rephase the paragraph.

Line 228, what does BW in Table 1 under Income category refer to?

Line 242, it is suggested to present a PCA plot of Table 2 to better visualize how food groups are clustered according to the dietary patterns.

Line 249-254, please explain the biological meaning of % variance regarding the dietary pattern in each trimester.

Line 258, What method was used for analyzing the association with maternal characteristics and the three patterns presented in Table 3? Do “mean” and “SE” in Table 3 refer to coefficient? Please clarify it.

Author Response

Please find below the point-by-point answers to reviewer´s remarks on our manuscript. Indeed, the observations made helped us to significantly improve our study. We hope that our works has, in its present format, has reached the standards of publication of a high quality journal such as Nutrients

Yours, faithfully

Silvia Saldiva

Reviewer 2 Report

Dear Authors,

The study presented to me for review appears to be well designed and described; however, it is not free of methodological limitations. 

The study conditions generally exclude BMI levels for pregnant women. The study should describe the basis for using BMI level as an independent variable.

It would be reasonable to set research hypotheses for the purpose.

The Materials and Methods chapter is well described. The paragraph following Figure 1 in line 112 should be described as '2.2 Characteristics of the Study Group'.

It is disturbing that the results were collected almost a decade ago, and as the researchers themselves write in recent years in Brazil there has been a sharp increase in the percentage of women with excessive body weight, a huge impact on this condition may also have been caused by the COVID-19 pandemic, which significantly affected the Brazilian population after all. Not to mention socio-cultural changes over the past decade. 

The literature needs to be updated; as PubMed reveals, a number of valuable items on the topic under study have been published in the last 5 years that are worth citing. 

It would be worthwhile to focus on the practical implications that arise from the study and describe them more broadly. 

The results need to be clarified. Also refrain from using abbreviations in the results 'DP', 'EGWG'.

In an observational cohort study, among other things, it is possible to analyze the causal effects of various risk factors in this case poor nutrition and excessive weight gain, so it is possible to determine relative risk which the researchers did quite well and for which I congratulate them. Despite the numerous limitations , the researchers tried to use all possible means to limit the researcher's error. 

Greetings!

Author Response

(The authors gave the same response as above.)
